# Divergent evolution of *Corynebacterium diphtheriae* in India: An update from National Diphtheria Surveillance network

**Naveen Kumar Devanga Ragupathi**[1,2☯], **Dhiviya Prabaa Muthuirulandi Sethuvel**[1☯], **Dhivya Murugan**[1], **Ranjini Ranjan**[1], **Vikas Gautam**[3], **Prashanth Gupta**[4], **Jaichand Johnson**[5], **Naresh Chand Sharma**[6], **Ankur Mutreja**[7,8], **Pradeep Haldar**[9], **Arun Kumar**[10], **Pankaj Bhatnagar**[10], **Lucky Sangal**[10], **Balaji Veeraraghavan**[1]*

1 Department of Clinical Microbiology, Christian Medical College, Vellore, India, 2 Department of Chemical and Biological Engineering, The University of Sheffield, Sheffield, United Kingdom, 3 Department of Medical Microbiology, Postgraduate Institute of Medical Education & Research, Chandigarh, India, 4 King George's Medical University, Lucknow, Uttar Pradesh, India, 5 State Public Health Laboratory, Thiruvananthapuram, India, 6 Maharishi Valmiki Infectious Diseases Hospital, New Delhi, India, 7 Department of Medicine, Addenbrookes Hospital, University of Cambridge, Cambridge, United Kingdom, 8 Translational Health Science and Technology Institute (THSTI), Delhi-NCR, India, 9 Ministry of Health and Family Welfare, Government of India, New Delhi, India, 10 World Health Organization, New Delhi, India

☯ These authors contributed equally to this work.
* vbalaji@cmcvellore.ac.in

**Data Availability Statement:** All relevant data are within the manuscript and its supporting information files.

## Abstract

Diphtheria is caused by a toxigenic bacterium *Corynebacterium diphtheria* which is being an emerging pathogen in India. Since diphtheria morbidity and mortality continues to be high in the country, the present study aimed to study the molecular epidemiology of *C. diphtheriae* strains from India. A total of 441 diphtheria suspected specimens collected as part of the surveillance programme between 2015 and 2020 were studied. All the isolates were confirmed as *C. diphtheriae* with standard biochemical tests, ELEK's test, and real-time PCR. Antimicrobial susceptibility testing for the subset of isolates showed intermediate susceptibility to penicillin and complete susceptible to erythromycin and cefotaxime. Isolates were characterized using multi locus sequence typing method. MLST analysis for the 216 *C. diphtheriae* isolates revealed major diversity among the sequence types. A total of 34 STs were assigned with majority of the isolates belonged to ST466 (30%). The second most common ST identified was ST405 that was present in 14% of the isolates. The international clone ST50 was also seen. The identified STs were grouped into 8 different clonal complexes (CC). The majority belongs to CC5 followed by CC466, CC574 and CC209, however a single non-toxigenic strain belongs to CC42. This epidemiological analysis revealed the emergence of novel STs and the clones with better dissemination properties. This study has also provided information on the circulating strains of *C. diphtheriae* among the different regions of India. The molecular data generated through surveillance system can be utilized for further actions in concern.

**Funding:** The study was funded by World Health Organization, Country Office, New Delhi, India (Ref. No: 2014/488695).

**Competing interests:** The authors declare that no competing interest exist.

## 1. Introduction

Diphtheria is a vaccine preventable disease caused by the human pathogen toxigenic strains of *Corynebacterium diphtheria*e, which causes severe infections like myocarditis and peripheral neuropathy. Humans are the only known reservoir of *C. diphtheriae* [1]. Whereas the other two toxin producing *Corynebacterium* species such as *Corynebacterium ulcerans* and *Corynebacterium pseudotuberculosis* are of zoonotic origin. Non-toxigenic strains of *C. diphtheriae* can also cause disease with less severity. Diphtheria, if not detected early and treated promptly can lead to significant mortality and morbidity [2]. According to WHO, India reported the highest number of diphtheria cases with more than half of the overall global diphtheria cases between 1980 and 2019 [3].

Recently, effective vaccination with diphtheria toxoid has reduced the mortality and morbidity of diphtheria infection globally. However, fully vaccinated individuals also get affected for unknown reasons including waning immunity, personal health, nutritional status, and infection due to non-toxigenic or variant strains [1]. In 2014, the Government of India launched "Mission Indradhanush" to strengthen the immunization programme and to rapidly achieve full immunization coverage of all children and pregnant women against seven vaccine preventable diseases including diphtheria. In 2015, WHO India also initiated a national surveillance programme on vaccine preventable diseases. In which, Christian Medical College, Vellore, acts as a National reference laboratory (NRL) for diphtheria and pertussis surveillance.

Outbreaks associated with displaced population, socioeconomic conditions of the people and vaccination rates emphasize the continued threat posed by diphtheria. Sporadic outbreaks of diphtheria have continued to be reported from different parts of India among both vaccinated and non-vaccinated individuals [2, 4, 5]. Information on the clonality of *C. diphtheriae* strains is not well established in India. Previous MLST studies have shown the dominance of certain clones in a specific geographical area [1]. Preliminary investigations in India have reported novel sequence types in most places including the outbreak reported in Northern Kerala in 2016 [6, 7]. This investigation shows that the genome of *C. diphtheriae* is constantly evolving and necessitates the monitoring of the spread and evolutional changes in endemic region like India. Owing to the continuous diphtheria outbreak in the country, the present study investigated the molecular epidemiology of geographically diverse collection of *C. diphtheriae* strains from India.

## 2. Results

### 2.1. Culture, AST and real-time PCR

A total of 441 diphtheria suspected specimens [isolates ($n = 204$); throat swabs ($n = 218$); other samples ($n = 19$)] were received and screened for diphtheria by both culture and real-time PCR. These included samples collected at study participating centres. All these samples were confirmed at the NRL for diphtheria at CMC, Vellore. Detailed breakdown of positives by culture and PCR were given in S1 Table. Out of 218 swabs received, 32 were culture positive and 43 were PCR positive for *C. diphtheriae*. One isolate was identified to be toxin (*tox*A) negative by PCR. Of the 204 isolates received from participating centres, 180 were confirmed as *C. diphtheriae* by PCR with 8.3% ($n = 15$) *tox*A negative isolates. Among the 19 other samples, 10 were found to be positive, of which 7 were toxin negative. Antimicrobial susceptibility testing for the subset of 33 isolates (CMC) showed intermediate susceptibility to penicillin. Whereas all tested isolates were susceptible to erythromycin and cefotaxime according to CLSI-M45 guidelines.

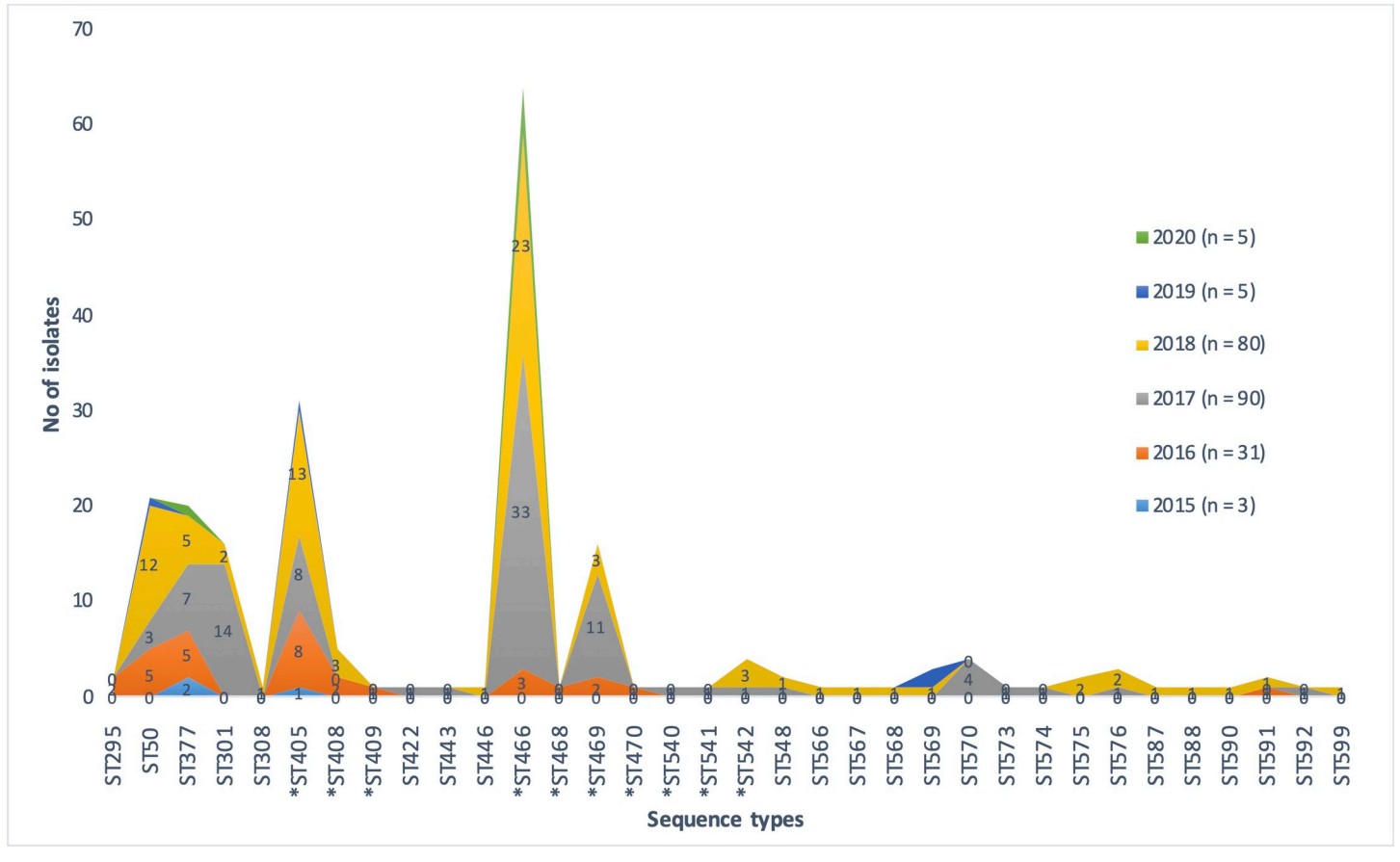

**Fig 1. Distribution of sequence types over the study period (2015–2020).**

## 2.2. MLST analysis

MLST analysis for the 216 *C. diphtheriae* isolates representing the Indian population revealed major diversity among the sequence types. A total of 34 STs were assigned to the 216 *C. diphtheria* isolates. The distribution of STs over the study period was shown in Fig 1. The identified STs were grouped into 8 different clonal complexes (CC). The majority belongs to CC5 followed by CC466, CC574 and CC209, however a single non-toxigenic strain belongs to CC42.

Majority of the isolates belonged to ST466 (30%). The second most common ST identified was ST405 that was present in 14% of the isolates. The international clone ST50 was seen in 11% of the study isolates. Further ST377 and ST469 were identified in 9.2% and 7.4% of the isolates, respectively.

**2.2.1. Region specific clones.** ST301 was identified only in Lucknow and was not seen in any other state. The non-toxigenic clone was mainly seen in the Kerala isolates. The region-wide distribution of observed *C. diphtheriae* STs is given in S2 Table. Other STs observed were ST295, ST308 including novel STs such as ST408, ST409, ST422, ST443, ST446, ST468, ST470, ST540, ST541, ST542, ST548, ST566, ST567, ST568, ST569, ST57, ST573, ST574, ST575, ST576, ST587, ST588, ST590, ST591, ST592 and ST599.

Most of the novel STs were identified in isolates from Kerala. All the novel STs identified in this study were SLV/DLV/TLV of the existing STs with a minimum of one SNP to a maximum of 26 SNPs except for two STs (ST466 and ST566), which stand separate (Table 1). The non-toxigenic isolates were observed to have the following sequence types ST542, ST409, ST295,

**Table 1. SNP analysis of sequence types identified among the study isolates.**

| Sequence type | atpA | dnaE | dnaK | fusA | leuA | odhA | rpoB | SNPs | Clonal complex |
|---|---|---|---|---|---|---|---|---|---|
| ST3 | 3 | 3 | 3 | 2 | 3 | 3 | 3 | Known ST | CC209 |
| ST295 | 2 | 1 | 20 | 19 | 3 | 3 | 14 | Known ST | CC509 |
| ST50 | 2 | 2 | 4 | 1 | 3 | 3 | 2 | Known ST | CC5 |
| ST377 | 2 | 43 | 87 | 4 | 59 | 3 | 6 | Known ST | CC540 |
| ST301 | 2 | 10 | 3 | 1 | 7 | 3 | 2 | Known ST | CC574 |
| ST308 | 4 | 12 | 4 | 1 | 18 | 3 | 13 | Known ST | CC5 |
| *ST405 | 4 | 12 | 4 | 1 | 18 | 3 | 2 | SLV of ST308 by 3 SNPs | CC5 |
| *ST408 | 8 | 2 | 3 | 1 | 18 | 3 | 13 | TLV of ST308 by atpA—4 dnaE—5 dnaK—11 SNPs | |
| *ST409 | 6 | 7 | 21 | 12 | 9 | 12 | 11 | SLV of ST163 by 1 SNP | CC42 |
| ST422 | 2 | 2 | 38 | 58 | 3 | 3 | 22 | TLV of ST50 by dnaK—17 fusA—6 rpoB—3 SNPs | |
| ST443 | 3 | 1 | 20 | 19 | 37 | 3 | 35 | TLV of ST295 by atpA—2 leuA—4 rpoB—1 SNPs | |
| ST446 | 4 | 2 | 4 | 1 | 3 | 3 | 2 | SLV of ST50 by 4 SNPs | CC5 |
| *ST466 | 2 | 3 | 3 | 1 | 2 | 3 | 13 | None | CC466 |
| *ST468 | 3 | 2 | 4 | 19 | 59 | 3 | 13 | DLV of ST136 by fusA—7 leuA—4 | CC5 |
| *ST469 | 4 | 10 | 3 | 1 | 7 | 3 | 13 | DLV of ST301 by atpA—4 rpoB—3 SNPs | |
| *ST470 | 3 | 2 | 99 | 1 | 20 | 3 | 13 | TLV of ST408 by atpA—2 dnaK—7 leuA—6 SNPs | |
| *ST540 | 2 | 43 | 87 | 4 | 2 | 3 | 6 | SLV of ST377 by 8 SNPs | CC377 |
| *ST541 | 2 | 3 | 3 | 1 | 59 | 3 | 13 | SLV of ST466 by 8 SNPs | CC466 |
| *ST542 | 3 | 1 | 4 | 19 | 5 | 3 | 9 | TLV of ST468 by dnaE—1 leuA—3 rpoB—3 SNPs | |
| ST548 | 3 | 3 | 4 | 2 | 3 | 3 | 5 | SLV of ST209 by 4 SNPs | CC209 |
| ST566 | 19 | 9 | 4 | 19 | 30 | 26 | 22 | None | |
| ST567 | 23 | 3 | 3 | 1 | 2 | 3 | 13 | SLV of ST466 by 1 SNP | CC466 |
| ST568 | 3 | 1 | 49 | 19 | 3 | 3 | 6 | TLV of ST542 by dnaK—10 leuA—1 rpoB—3 SNPs | |
| ST569 | 3 | 2 | 4 | 19 | 3 | 3 | 13 | SLV of ST468 by 4 SNPs | CC5 |
| ST570 | 3 | 3 | 3 | 2 | 41 | 3 | 3 | SLV of ST3 by 2 SNPs | CC209 |
| ST573 | 2 | 3 | 76 | 1 | 25 | 3 | 13 | DLV of ST466 by dnaK—5 leuA—3 SNPs | CC466 |
| ST574 | 2 | 10 | 3 | 1 | 3 | 3 | 2 | SLV of ST301 by 6 SNPs | CC574 |
| ST575 | 3 | 1 | 4 | 58 | 5 | 3 | 22 | DLV of ST542 by fusA -1 rpoB—5 SNPs | |
| ST576 | 2 | 10 | 3 | 64 | 7 | 3 | 2 | SLV of ST301 by 1 SNP | CC574 |
| ST587 | 2 | 3 | 97 | 1 | 2 | 3 | 35 | DLV of ST466 dnaK—8 rpoB—1 SNPs | CC466 |
| ST588 | 2 | 3 | 3 | 1 | 2 | 16 | 13 | SLV of ST466 by odhA—4 SNP | CC466 |
| ST590 | 2 | 10 | 3 | 1 | 3 | 57 | 2 | DLV of ST301 by leuA—6 odhA—1 SNPs | CC574 |
| ST591 | 2 | 3 | 38 | 1 | 3 | 3 | 22 | DLV of ST422 by dnaE—1 fusA—6 SNPs | |
| ST592 | 2 | 12 | 87 | 4 | 59 | 3 | 2 | DLV of ST377 by dnaE—4 rpoB—1 SNPs | |
| ST599 | 2 | 12 | 4 | 1 | 18 | 3 | 2 | SLV of ST405 by atpA—4 SNP | CC5 |

ST443, ST422, ST566, and ST568, whereas the STs identified in both toxigenic and non-toxigenic isolates were ST408, ST466, ST591 and ST575, respectively. The MLST sequences obtained, and its distribution among the study isolates are given in the S2 Table. The phylogenetic tree showed the relatedness of the isolates based on the concatenated MLST gene sequences (Fig 2). Individual isolate details on toxigenicity, ST, region and year of isolation were given in S3 Table. The possible routes of transmission of STs were mapped according to their pattern of first occurrence and their sample collection date (Fig 3).

## 3. Discussion

Diphtheria remains endemic in many parts of the world and continues to be a major public health problem in India. There have been numerous reports of diphtheria from several parts of

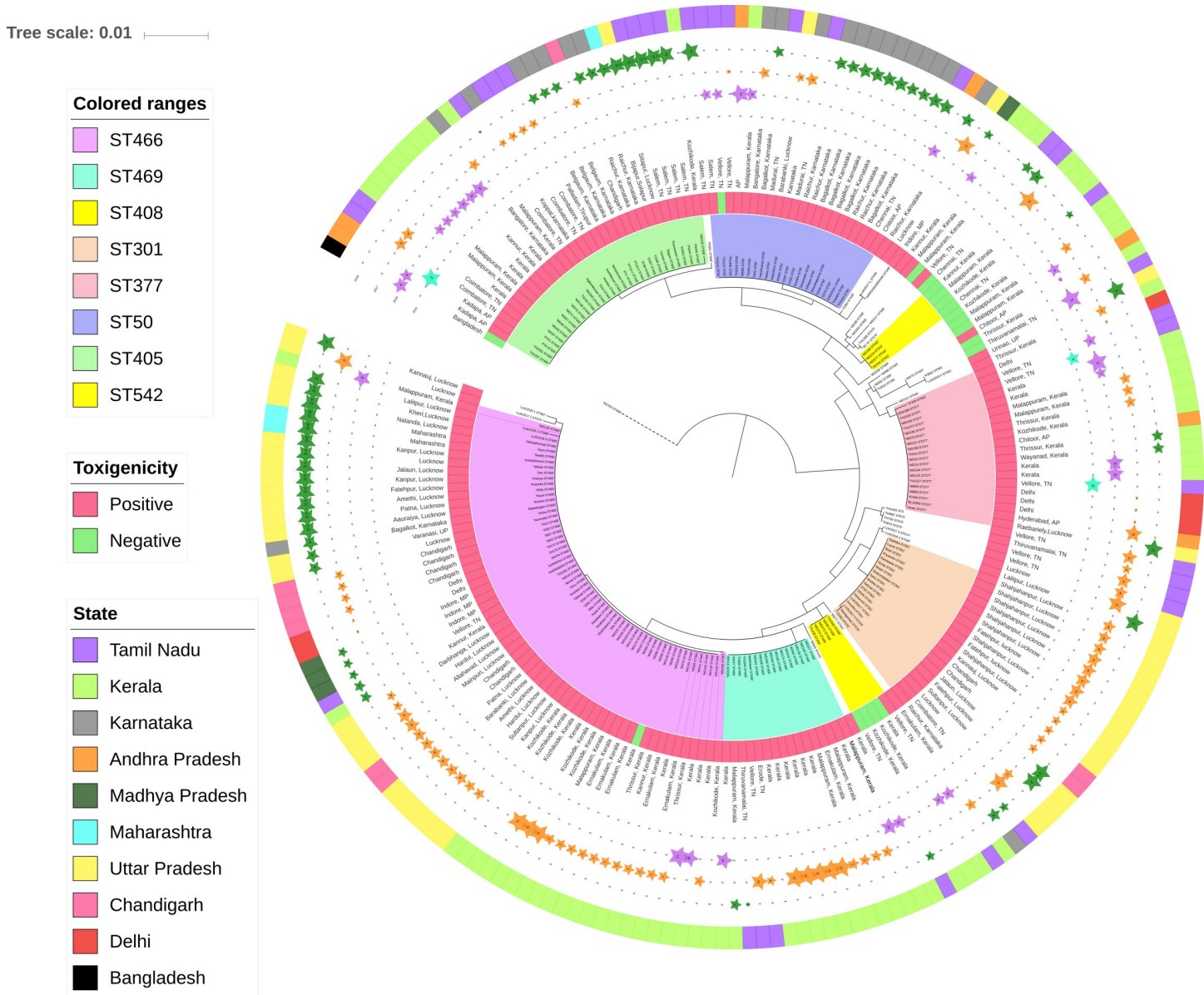

**Fig 2. Maximum likelihood phylogeny of MLST alleles from 216 *C. diphtheriae* isolates.** The clades are highlighted based on the common ST types identified in the study. STs associated with non-toxigenic strains are highlighted in yellow. The inner ring represents the toxigenicity results of the isolates followed by date of collection. The outer ring represents the region of isolation.

India for the past 10 years [8]. The occurrence of novel sequence types and their circulation in different geographical regions has recently become a serious concern. To the best of our knowledge, this is the first extensive study of the epidemiology of *C. diphtheriae* from India.

Amongst the total *C. diphtheriae* positive cases in this study, most of them were from Tamil Nadu, Kerala and Uttar Pradesh (Lucknow) which indicates the burden of diphtheria in the respective localities. Epidemiological surveillance provides significant understanding of the clonal population of *C. diphtheriae* particularly in endemic region like India. The presented MLST analysis of the *C. diphtheriae* isolates revealed great diversity among the isolates over the years 2015–2020. The MLST analysis also includes isolates from the Kerala outbreak reported in 2016 [6].

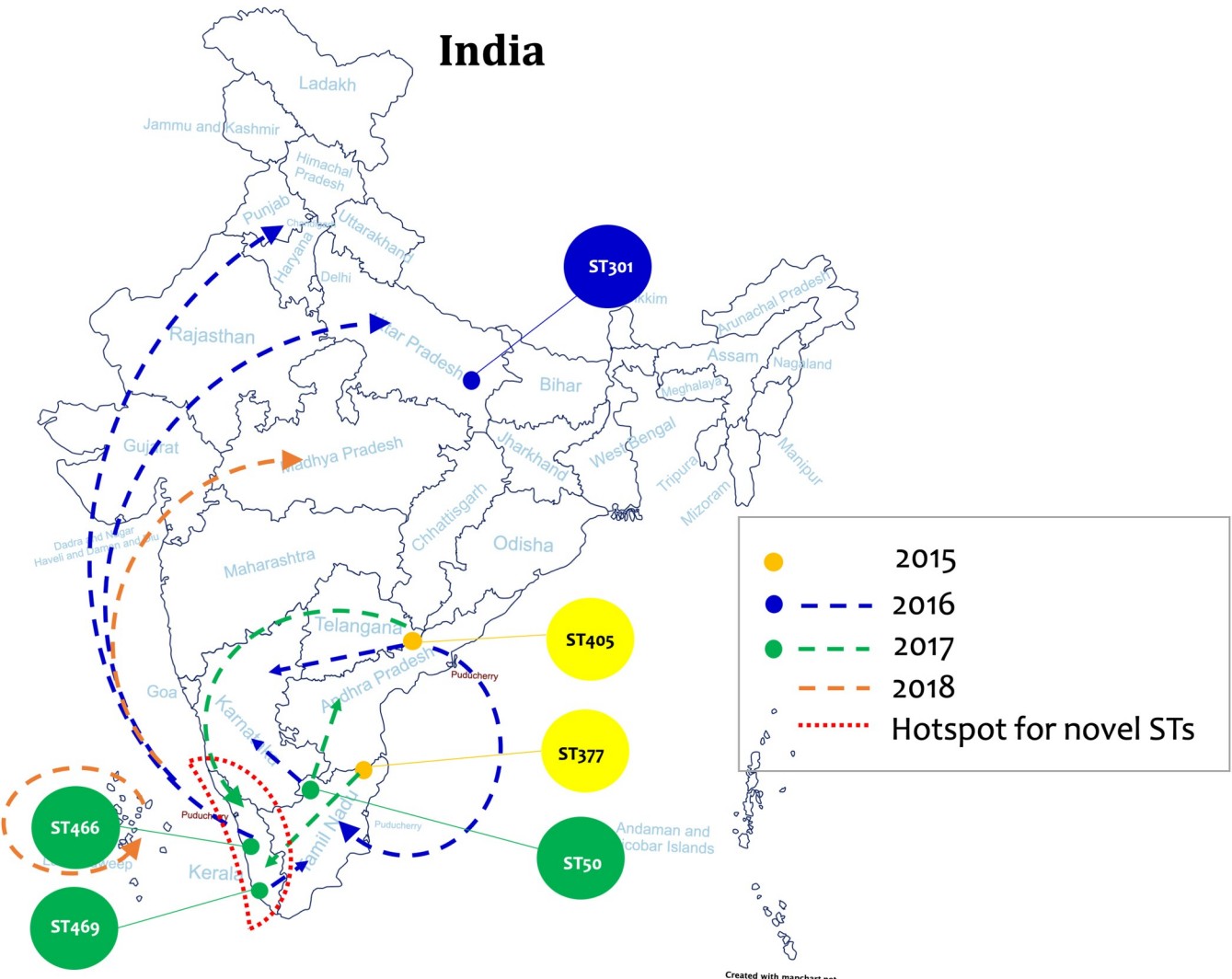

**Fig 3. Illustration of the probable routes of transmission of STs identified in this study.** The coloured circles represent the first occurrence of the ST and the dotted arrow with colour indicates the possible transmission routes according to the year of isolation. The red dotted lines highlight the hotspot for novel sequence types. Map outline was created using mapchart.net. Republished from mapchart.net under a CC BY license, with permission from MapChart, original copyright 2021.

The predominant sequence type in this study was ST466 (28.6%), a novel sequence type which was first observed in the Kerala outbreak in 2016. Later in 2017, the ST was consistently seen in Kerala and was also found in Lucknow and Chandigarh. While in 2018, it was found in significant numbers in Lucknow, Uttar Pradesh and Madhya Pradesh which shows the ability of this clone to disseminate rapidly. The second most common sequence type was ST405 (14.5%). This is an SLV of ST308 and was first observed in the year 2015 in Andhra Pradesh. This clone further spread to Kerala in 2016. While in 2017, it was also seen in Tamil Nadu and Karnataka. However, in 2018, ST405 was found endemic in Karnataka. Notably, all the identified novel sequence types evolved from established known sequence types by locus variations.

ST377 (9.2%) which was identified among isolates from Tamil Nadu in 2015, began to be seen in Kerala after 2016. Whereas ST469 (7.8%) was first identified in Kerala in 2016 and from 2017 it was also seen in Tamil Nadu.

Interestingly, a region-specific sequence type was identified in this study. ST301 which was restricted to Lucknow was not found in any other region in India. ST301 was first identified in 2017 and continue to persist in Lucknow. A SLV of ST301 was identified and assigned the novel type as ST576. This shows region specific emergence of novel clones due to SNPs in the existing clones.

Results from this study reveal that *C. diphtheriae* strains are highly diverse. Till date different STs have reported from different countries [9–11]. Such global outbreak associated clones include, ST8, ST12, ST52, and ST66 from United Kingdom [9]. Similarly, ST378 and ST395 seems to be common in South Africa [11]. Whereas, in India ST405, ST466 and ST377 are common as observed in this study. Moreover, the international clone, ST50 was identified in 2016 in Tamil Nadu and Andhra Pradesh. In 2017 and 2018 it was common in Karnataka which shows the inter-state spread of the ST50 clone. Most of the STs identified in India are diverse but the reason for this and the source of these clones is not well established. Socio-geographical approach is required to understand and identify the ancestral lineage of these clones.

It has been postulated that non-toxigenic strains occur more frequently in individuals who have been previously immunized [12]. In this study, though the immunization status was not known, 14% of strains were negative for toxin by PCR. Further, phylogenetic analysis revealed that majority of the non-toxigenic strains were clustered in two different clades associated with ST408 and ST542. These were mostly seen in isolates from Kerala. Non-toxigenic ST408 and ST542 were found to have evolved from the existing sequence types of toxigenic ST301 and ST377 respectively. This scenario could be due to the loss of genes that affect the expression of *tox* gene during the evolution.

MLST is a useful tool for evolutionary studies and for tracking the dissemination of clones. However, there is a limitation in discriminating strains within the same ST due to continuing change in their accessory genomes. Previous studies revealed that approximately one-third of the *C. diphtheriae* genome encodes accessory genes that vary widely between strains. The strains within individual STs differed by the presence or absence of up to 290 genes, which are mostly present on the genomic islands [13]. This study clearly indicates the evolving genome diversity of *C. diphtheriae* strains in India.

Clonal variation among the *C. diphtheriae* strains was observed in the study population. The epidemiological analysis revealed the emergence of novel STs and the clones with better dissemination properties. This indicates a rapid evolution of *C. diphtheriae* in India. In the present study, ST466 was found to be the predominant sequence type with better adaptive and dissemination capabilities. ST301 was identified as a region-specific toxigenic clone, from which the non-toxigenic ST408 has evolved. This study has now revealed the pattern of circulating STs of *C. diphtheriae* among the different regions of India.

## 4. Materials and methods

Samples collected as part of the National Diphtheria Surveillance Programme, India were included in this study. Specimens (throat, ear and nasal swabs, and pieces of throat membrane) were collected from diphtheria suspected patients attending CMC, Vellore. Isolates were also received from participating centres, various district hospitals and government health centres. All the specimens were subjected to culture identification and were confirmed by real-time PCR.

The collected samples were representative of the Indian population. The centre from South India includes, Christian Medical College, Vellore; Coimbatore Medical College, Coimbatore; Madurai Medical College, Madurai; Kauvery Hospital, Trichy; Govt. Villupuram Medical College, Villupuram; Kanchi Kamakoti Hospital and Southern Railway Hospital, Chennai; State

Public Health, Trivandrum, Kerala; St. John's Hospital, RIMS, Raichur and S. Nijalingappa Medical College, Karnataka. The samples representing the North Indian population were collected from the following centres: PGIMER, Chandigarh; KGMC, Lucknow; Bharathi Vidya Peeth, Sangli, Maharashtra; and LTM Medical College, Mumbai.

### 4.1. Bacterial culture and characterisation

The swabs were processed, and initial identification was done by staining and culture on blood agar (10% sheep blood) and serum tellurite agar. *C. diphtheriae* colonies were further confirmed on tinsdale agar (cystinase test) (DIFCO, USA) as black colonies with brown halo. The species identification was done with standard biochemical testing based on the utilization of sugars such as glucose, dextrose, sucrose, maltose followed by nitrate and urease tests (Fisher Scientific, Massachusetts, USA).

### 4.2. Antimicrobial susceptibility testing

Minimum inhibitory concentration (MIC) was determined for the subset of the isolates using E-test against antibiotics such as penicillin, cefotaxime, and erythromycin. The results were interpreted according to CLSI-M45 guidelines [14].

### 4.3. Real-time PCR detection

DNA was extracted using the AQIAamp DNA blood mini kit (QIAGEN, Hilden, Germany). The target genes include *rpo*B specific for *C. diphtheria* and *C. ulcerans*, *tox*A. RNaseP was used as an internal control. Primers, probes and cyclic conditions used in this study were as previously described [15]. Briefly, the reaction set-up includes 95˚C for 10 mins; followed by 45 cycles of 95˚C for 15 secs and 60˚C for 30 secs. Ct cut off values for positivity were ≤34 for *C. diphtheriae*, ≤31 for *C. ulcerans*, and ≤35 for *tox*A.

### 4.4. MLST typing

*C. diphtheriae* isolates were subjected to multi-locus sequence typing (MLST) according to the *C. diphtheriae* PubMLST protocol. Briefly, the DNA fragments on each strand were sequenced with the sequencing primers by using an ABI PRISM BigDye Terminator Cycle Sequencing kit (Applied Biosystems, CA, USA) and an ABI 3500 Genetic Analyser (Applied Biosystems, USA). The Allele profiles, ST and clonal complexes (CC) were assigned by using the PubMLST database (https://pubmlst.org/organisms/corynebacterium-diphtheriae) [9]. Alleles and ST that had not been previously described were submitted to the database and were assigned new allele numbers and STs. A clonal complex was defined as a cluster of related STs linked as single-, double- or triple- locus variants to another ST in the group. MLST alleles of *C. diphtheriae* isolates were used to build a phylogeny based on the maximum-likelihood method using MEGA (v7.0.26) [16]. Metadata were added to the phylogenetic tree using iTOL (v4.3) online software [17]. Further, predicted routes of transmission of strains studied here was mapped using mapchart.net.

### 4.5. goeBURST analysis

goeBURST analysis was performed to identify the related genotypes in the population, and to identify the founding genotype (ST) of each group. goeBURST analysis was performed using the PHYLOViZ 2.0 tool [18].

## Supporting information

**S1 Table. Laboratory confirmation of *C. diphtheriae* cases at NRL for diphtheria, CMC, Vellore.**
(DOCX)

**S2 Table. Distribution of sequence types among different states in India.**
(DOCX)

**S3 Table. Metadata including isolate ID, toxigenicity, STs, region and year of isolation for *C. diphtheriae* isolates depicted in the MLST phylogeny Fig 2.**
(XLSX)

## Acknowledgments

The authors gratefully acknowledge the Institutional Review Board of the Christian Medical College, Vellore (83-i/11/13) for approving the study and providing lab space and facilities.

## Author Contributions

**Conceptualization:** Naveen Kumar Devanga Ragupathi, Dhiviya Prabaa Muthuirulandi Sethuvel, Lucky Sangal, Balaji Veeraraghavan.

**Data curation:** Dhiviya Prabaa Muthuirulandi Sethuvel.

**Formal analysis:** Naveen Kumar Devanga Ragupathi, Vikas Gautam, Prashanth Gupta, Jaichand Johnson, Naresh Chand Sharma, Ankur Mutreja, Arun Kumar, Pankaj Bhatnagar, Balaji Veeraraghavan.

**Funding acquisition:** Pradeep Haldar, Arun Kumar, Pankaj Bhatnagar, Lucky Sangal.

**Investigation:** Naveen Kumar Devanga Ragupathi, Vikas Gautam, Prashanth Gupta, Jaichand Johnson, Naresh Chand Sharma.

**Methodology:** Dhivya Murugan, Ranjini Ranjan.

**Resources:** Vikas Gautam, Prashanth Gupta, Jaichand Johnson, Naresh Chand Sharma, Pradeep Haldar, Arun Kumar, Pankaj Bhatnagar, Lucky Sangal.

**Software:** Naveen Kumar Devanga Ragupathi.

**Supervision:** Naveen Kumar Devanga Ragupathi, Balaji Veeraraghavan.

**Validation:** Naveen Kumar Devanga Ragupathi, Pradeep Haldar, Pankaj Bhatnagar, Lucky Sangal, Balaji Veeraraghavan.

**Writing – original draft:** Naveen Kumar Devanga Ragupathi, Dhiviya Prabaa Muthuirulandi Sethuvel, Lucky Sangal, Balaji Veeraraghavan.

**Writing – review & editing:** Ankur Mutreja, Pradeep Haldar, Arun Kumar, Pankaj Bhatnagar, Lucky Sangal, Balaji Veeraraghavan.

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
