## [Decision Letter · Decision Letter 0]

20 Sep 2021

PONE-D-21-24753Divergent evolution of Corynebacterium diphtheriae in India: An update from National Diphtheria Surveillance NetworkPLOS ONE

Dear Dr. Veeraraghavan,

Thank you for submitting your manuscript to PLOS ONE. After careful consideration, we feel that it has merit but does not fully meet PLOS ONE’s publication criteria as it currently stands. Therefore, we invite you to submit a revised version of the manuscript that addresses the points raised during the review process. Your manuscript has been reviewed and a minor revision  is suggested.  Please follow the comments and make the necessary revision. If you don't agree, you can give a rebuttal.   

We look forward to receiving your revised manuscript.

Kind regards,

Yung-Fu Chang

Academic Editor

PLOS ONE

Journal Requirements:

3.Please update your submission to use the PLOS LaTeX template. The template and more information on our requirements for LaTeX submissions can be found at http://journals.plos.org/plosone/s/latex.

4. We note that Figure 3 in your submission contain map images which may be copyrighted. All PLOS content is published under the Creative Commons Attribution License (CC BY 4.0), which means that the manuscript, images, and Supporting Information files will be freely available online, and any third party is permitted to access, download, copy, distribute, and use these materials in any way, even commercially, with proper attribution. For these reasons, we cannot publish previously copyrighted maps or satellite images created using proprietary data, such as Google software (Google Maps, Street View, and Earth). For more information, see our copyright guidelines: http://journals.plos.org/plosone/s/licenses-and-copyright.

1. You may seek permission from the original copyright holder of Figure 3 to publish the content specifically under the CC BY 4.0 license.  

Reviewers' comments:

Reviewer's Responses to Questions

**Comments to the Author**

1. Is the manuscript technically sound, and do the data support the conclusions?

Reviewer #1: Yes

2. Has the statistical analysis been performed appropriately and rigorously? 

Reviewer #1: Yes

3. Have the authors made all data underlying the findings in their manuscript fully available?

Reviewer #1: No

4. Is the manuscript presented in an intelligible fashion and written in standard English?

Reviewer #1: Yes

5. Review Comments to the Author

Reviewer #1: Kumar et al., describe an important study on 441 which suggests that novel STs and clones are emerging in India with enhanced ability to disseminate and also provides a very useful snapshot of the strains of C diphtheriae circulating in India. This is an important study.

I have a few comments to the authors

the data statement says all the data is available, but I cannot find links to the MLST profiles or teh data used in the iTOL tree....

How related are the strains described in this manuscript to those in Will et al 2021 (Nat Comms) there is overlap in the authorship and it is important that the strains with whole genomes can be cross referenced – this must be stated.

It would be good to have more international context too….such as relationships to other Asian strains and the reference to global strains in the text (around the text lines 244-253) – is it possible to add some date context to these…and potentially speculate on the emergence of these in India? Where they introduced to india? Or was india the source of these clones globally….

Line 63 – ‘India reported the highest number of diphtheria cases 64 globally’

…this sentence should be contextualised either with a date ‘India reported the highest number of diphtheria cases 64 globally between 2015 and 2020’ [if that is correct?]

or made open ended ‘India reports the highest number of diphtheria cases 64 globally’

line 97/98 the unknown reason for why vaccinated individuals become affected…such as waning immunity or strain variation…

line 318 – please provide PubMLST link and this also probably needs the link to Bolt et al for the primers references…

6. PLOS authors have the option to publish the peer review history of their article (what does this mean?). If published, this will include your full peer review and any attached files.

Reviewer #1: No

---

## [Author Response · Author response to Decision Letter 0]

18 Nov 2021

1. the data statement says all the data is available, but I cannot find links to the MLST profiles or teh data used in the iTOL tree....

Data for all individual isolates included in the iTOL tree are now given in supplementary table 3. Line 161-162.

2. How related are the strains described in this manuscript to those in Will et al 2021 (Nat Comms) there is overlap in the authorship and it is important that the strains with whole genomes can be cross referenced – this must be stated.

The current study does not represent whole genome data and it primarily discusses the epidemiological stand of C. diphtheriae in India. However, citation to will et al is included to refer detailed information on few isolates from this study included in study by Will et al.

3. It would be good to have more international context too….such as relationships to other Asian strains and the reference to global strains in the text (around the text lines 244-253) – is it possible to add some date context to these…and potentially speculate on the emergence of these in India? Where they introduced to india? Or was india the source of these clones globally….

More than 70% of the isolates observed in India are found to be novel STs and are not reported elsewhere. MLST data is not sufficient to speculate origin of existing STs. It needs complete genome information from global data set to compare with the Indian isolates, which itself is a separate study. However, to provide a global representation, previous information on existing STs observed in India is included in the discussion as suggested by the reviewer.

Line 63 – ‘India reported the highest number of diphtheria cases 64 globally’

…this sentence should be contextualised either with a date ‘India reported the highest number of diphtheria cases 64 globally between 2015 and 2020’ [if that is correct?]

or made open ended ‘India reports the highest number of diphtheria cases 64 globally’

The statement has been modified to include details on the number of cases reported as suggested. “India reported the highest number of diphtheria cases globally as of October 2020”.

line 97/98 the unknown reason for why vaccinated individuals become affected…such as waning immunity or strain variation…

The sentence has been modified as “However, fully vaccinated individuals also get affected for unknown reasons including waning immunity, personal health, nutrition status, and infection due to non-toxigenic or variant strains”. Line 97-99.

line 318 – please provide PubMLST link and this also probably needs the link to Bolt et al for the primers references…

 The link for PubMLST is now updated and included reference for Bolt et al. 2010 as suggested (Line 283).

---

## [Decision Letter · Decision Letter 1]

22 Nov 2021

PONE-D-21-24753R1Divergent evolution of Corynebacterium diphtheriae in India: An update from National Diphtheria Surveillance NetworkPLOS ONE

Dear Dr. Veeraraghavan,

Thank you for submitting your manuscript to PLOS ONE. After careful consideration, we feel that it has merit but does not fully meet PLOS ONE’s publication criteria as it currently stands. Therefore, we invite you to submit a revised version of the manuscript that addresses the points raised during the review process. Your manuscript has been returned to the original reviewer and a minor revision is still suggested. 

Please submit your revised manuscript by two weeks If you will need more time than this to complete your revisions, please reply to this message or contact the journal office at plosone@plos.org. Please include the following items when submitting your revised manuscript:A rebuttal letter that responds to each point raised by the academic editor and reviewer(s). You should upload this letter as a separate file labeled 'Response to Reviewers'.A marked-up copy of your manuscript that highlights changes made to the original version. You should upload this as a separate file labeled 'Revised Manuscript with Track Changes'.An unmarked version of your revised paper without tracked changes. You should upload this as a separate file labeled 'Manuscript'.If applicable, we recommend that you deposit your laboratory protocols in protocols.io to enhance the reproducibility of your results. Protocols.io assigns your protocol its own identifier (DOI) so that it can be cited independently in the future. For instructions see: https://journals.plos.org/plosone/s/submission-guidelines#loc-laboratory-protocols. Additionally, PLOS ONE offers an option for publishing peer-reviewed Lab Protocol articles, which describe protocols hosted on protocols.io. Read more information on sharing protocols at https://plos.org/protocols?utm_medium=editorial-email&utm_source=authorletters&utm_campaign=protocols.

We look forward to receiving your revised manuscript.

Kind regards,

Yung-Fu Chang

Academic Editor

PLOS ONE

Journal Requirements:

Reviewers' comments:

Reviewer's Responses to Questions

**Comments to the Author**

1. If the authors have adequately addressed your comments raised in a previous round of review and you feel that this manuscript is now acceptable for publication, you may indicate that here to bypass the “Comments to the Author” section, enter your conflict of interest statement in the “Confidential to Editor” section, and submit your "Accept" recommendation.

Reviewer #1: (No Response)

2. Is the manuscript technically sound, and do the data support the conclusions?

Reviewer #1: Yes

3. Has the statistical analysis been performed appropriately and rigorously? 

Reviewer #1: Yes

4. Have the authors made all data underlying the findings in their manuscript fully available?

Reviewer #1: No

5. Is the manuscript presented in an intelligible fashion and written in standard English?

Reviewer #1: Yes

6. Review Comments to the Author

Reviewer #1: the authors have addressed most of the comments - however there are still a couple of things that are still not clear

“India reported the highest number of diphtheria cases globally as of October 2020” - this needs a date range and citation or further explanation...it currently reads like the most diphtheria cases globaly ever have occured in india up to october 2020 - it is important to state this as as it can be used to raise awareness and to obtain funding for the whole community.

Also the data files used to create the iTOL tree should be available as it will be difficult to look at in close detail once printed.

The overlap of strains in the Will et al., paper should also be explicitly stated as it will enable those genomes to be reanalysed

7. PLOS authors have the option to publish the peer review history of their article (what does this mean?). If published, this will include your full peer review and any attached files.

Reviewer #1: No

---

## [Author Response · Author response to Decision Letter 1]

26 Nov 2021

Response to reviewer comments

Reviewer #1: the authors have addressed most of the comments - however there are still a couple of things that are still not clear

“India reported the highest number of diphtheria cases globally as of October 2020” - this needs a date range and citation or further explanation...it currently reads like the most diphtheria cases globaly ever have occured in india up to october 2020 - it is important to state this as as it can be used to raise awareness and to obtain funding for the whole community.

 The statement has been modified as “According to WHO, India reported the highest number of diphtheria cases with more than half of the overall global diphtheria cases between 1980 and 2019”. Citation (Source: WHO vaccine-preventable diseases: monitoring system 2020 global summary) for the same is included in the introduction section (line 93-95).

Also the data files used to create the iTOL tree should be available as it will be difficult to look at in close detail once printed.

Data labelled in the figure 2 using iTOL is provided as the supplementary table 3.

The overlap of strains in the Will et al., paper should also be explicitly stated as it will enable those genomes to be reanalysed

 We thank the reviewer for pointing this. We have included the strain IDs in the supplementary table file S3. These isolate IDs can be used by the readers to refer corresponding genomes mentioned in Will et al., paper.

---

## [Decision Letter · Decision Letter 2]

2 Dec 2021

Divergent evolution of Corynebacterium diphtheriae in India: An update from National Diphtheria Surveillance Network

PONE-D-21-24753R2

Dear Dr. Veeraraghavan,

We’re pleased to inform you that your manuscript has been judged scientifically suitable for publication and will be formally accepted for publication once it meets all outstanding technical requirements.

Kind regards,

Yung-Fu Chang

Academic Editor

PLOS ONE

Additional Editor Comments (optional):

Reviewers' comments:

Reviewer's Responses to Questions

**Comments to the Author**

1. If the authors have adequately addressed your comments raised in a previous round of review and you feel that this manuscript is now acceptable for publication, you may indicate that here to bypass the “Comments to the Author” section, enter your conflict of interest statement in the “Confidential to Editor” section, and submit your "Accept" recommendation.

Reviewer #1: All comments have been addressed

2. Is the manuscript technically sound, and do the data support the conclusions?

Reviewer #1: Yes

3. Has the statistical analysis been performed appropriately and rigorously? 

Reviewer #1: Yes

4. Have the authors made all data underlying the findings in their manuscript fully available?

Reviewer #1: Yes

5. Is the manuscript presented in an intelligible fashion and written in standard English?

Reviewer #1: Yes

6. Review Comments to the Author

Reviewer #1: (No Response)

7. PLOS authors have the option to publish the peer review history of their article (what does this mean?). If published, this will include your full peer review and any attached files.

Reviewer #1: No

---

## [Editor Report · Acceptance letter]

6 Dec 2021

PONE-D-21-24753R2 

Divergent evolution of *Corynebacterium diphtheriae* in India: An update from National Diphtheria Surveillance Network 

Dear Dr. Veeraraghavan:

I'm pleased to inform you that your manuscript has been deemed suitable for publication in PLOS ONE. Congratulations! Your manuscript is now with our production department. 

Kind regards, 

on behalf of

Dr. Yung-Fu Chang 

Academic Editor

PLOS ONE